robotics/biomimetics/ocean engineering

aquaculture engineering, animal–robot interaction, biorobotics, bioinspired robotics

**Author for correspondence:**
M. Kruusmaa
e-mail: maarja.kruusmaa@taltech.ee

# Salmon behavioural response to robots in an aquaculture sea cage

M. Kruusmaa[1,2], R. Gkliva[2], J. A. Tuhtan[2], A. Tuvikene[3] and J. A. Alfredsen[1]

[1]Centre for Autonomous Marine Operations and Systems, Norwegian University of Science and Technology, Otto Nielsens Veg 10, Trondheim NO-7491, Norway
[2]Centre for Biorobotics, Department of Computer Systems, Tallinn University of Technology, Akadeemia tee 15A, 12618 Tallinn, Estonia
[3]Institute of Agricultural and Environmental Sciences, Estonian University of Life Sciences, Fr.R.Kreutzwald 5, 51006 Tartu, Estonia

MK, 0000-0001-5738-5421; JAT, 0000-0003-0832-7334

Animal–robot studies can inform us about animal behaviour and inspire advances in agriculture, environmental monitoring and animal health and welfare. Currently, experimental results on how fish are affected by the presence of underwater robots are largely limited to laboratory environments with few individuals and a focus on model species. Laboratory studies provide valuable insight, but their results are not necessarily generalizable to larger scales such as marine aquaculture. This paper examines the effects of underwater robots and a human diver in a large fish aggregation within a Norwegian aquaculture facility, with the explicit purpose to improve the use of underwater robots for fish observations. We observed aquaculture salmon's reaction to the flipper-propelled robot U-CAT in a sea cage with 188 000 individuals. A significant difference in fish behaviour was found using U-CAT when compared to a thruster-driven underwater robot, Argus Mini and a human diver. Specifically, salmon were more likely to swim closer to U-CAT at a lower tailbeat frequency. Fish reactions were not significantly different when considering motor noise or when U-CAT's colour was changed from yellow to silver. No difference was observed in the distance or tailbeat frequency as a response to thruster or flipper motion, when actuated and passively floating robots were compared. These results offer insight into how large aggregations of aquaculture salmon respond to underwater robots. Furthermore, the proposed underwater video processing workflow to assess fish's response to underwater robots is simple and reproducible. This work provides a practical method to study fish–robot interactions, which can lead to improved underwater robot designs to provide more affordable, scalable and effective solutions.

# 1. Introduction

The methods for investigating animal behaviour range from dummies and mirrors to audio and video playback and computer animations [1]. Robots can provide a new set of experimental methods because they are physically present, allowing a controlled variation of predesigned morphologies and movements [2,3]. Tools for animal–robot interactions are categorized in [4] as non-biomimetic, intra-species biomimetic (mimicking another species) or biomimetic (mimicking the same species). Examples include a mobile robot for herding ducks [5], a robotic honeybee for the analysis of dance communication [6], a multirobot system to investigate cockroach communal roosting [7], a robotic toy for evaluating canine anxiety [8] and a remotely controlled robotic car to quantify the chasing behaviour of dogs [9].

Fish have been traditionally studied using dummies, images and videos. The use of robotic models now permits investigating a wider set of sensory signatures [10–13]. In laboratory settings, the response of golden shiners to robot fish colour and tailbeat frequency has been investigated in [14]. Zebrafish's response to colour, aspect ratio, noise and tailbeat frequency has been studied in [15]. The response of mosquito fish to varying swimming depths and aspect ratios is reported in [16]. A robotic replica was used in a feedback loop to control fish behaviour [17], and in [18], the social responsiveness of a guppy is investigated using a biomimetic fish replica. In addition to a fish's physical response to biomimetic robots, investigators have also found that swarms of interacting fish robots can aid in studying fish foraging behaviour [19].

Laboratory conditions allow for controlled experiments with minimal environmental noise. Moreover, the use of model species such as zebrafish provide comparative studies with results found in literature. However, it is not self-evident that the results of the laboratory experiments are generalizable to animals in natural, noisy and uncontrollable environments, to different species (e.g. wild versus farm-raised) or to large aggregations of the same species [20].

In contrast with the growing number of laboratory investigations of fish's response to underwater robots, outdoor investigations remain sparse in the literature. Notable exceptions are the author's own study on the response of wild-caught mackerel (3000 individuals) under partially controlled conditions [21], where it was found that fish reacted significantly to the size and speed of a towed fish robot. Another outdoor study made use of a remotely operated fish-shaped robot in a coral reef, with the motivation to develop a robot for minimally invasive marine life observation [22]. During these experiments, wild fish did not appear to flee, even coming as close as 1 m. However, the study did not include any quantitative or comparative analyses.

This paper investigates large-scale fish responses to robots in the context of practical, underwater observations in fish farms. We are motivated by the need to develop technologies to monitor the rapid growth of aquaculture, whose production is on track to eclipse that of traditional fisheries [23,24]. Intensive commercial aquaculture brings with it substantial concerns regarding fish welfare [25]. Increased stress levels of compact fish schools may cause aggressive and competitive behaviours [26]. In addition to concerns of animal suffering, stress also makes fish receptive to diseases and parasites and may lower productivity. It has been confirmed in previous work that aquaculture procedures change fish swimming speed and other behavioural indicators [27]. Stress-induced behaviours, however, are difficult to observe. For example, the estimation of size and visual injuries is performed using steady cameras and biomass sensors. A detailed assessment of individuals and, in particular, the distribution of behaviours is currently not possible because intervention may cause fish to alter their behaviour. There is also a need to optimize feeding, as feed costs may take up as much as 50% of the fish production cycle [28]. Distribution of food intake along the depth of the cage is difficult to estimate as fish are usually fed from the surface and individuals compete for food. Undisturbed observations of fish would permit estimating how food is distributed along the whole population. Thus, new methods of *in-situ* observation are needed for behavioural studies in fish farms.

A fish farm can be viewed as a semi-controlled environment where some variables (e.g. size and number of fish, and species) are known. Environmental parameters such as temperature, lighting conditions, currents and waves vary and are not known and not controllable. It also requires robots that perform reliably in field conditions and over prolonged periods. For example, it may not be feasible to scale the robot down to the size of a fish because such a robot is not powerful enough to swim against currents in natural environments and cannot carry all the payload necessary for observations.

In this paper, we investigate the behaviour of fish in a semi-natural environment in a very large aggregation, in a commercial fish farm sea cage containing *ca* 190 000 individuals of farmed salmon. We have conducted experiments with U-CAT, a small flipper-propelled robot [29]. The initial hypothesis was that a flipper-propelled robot will offer a more suitable platform for fish observation

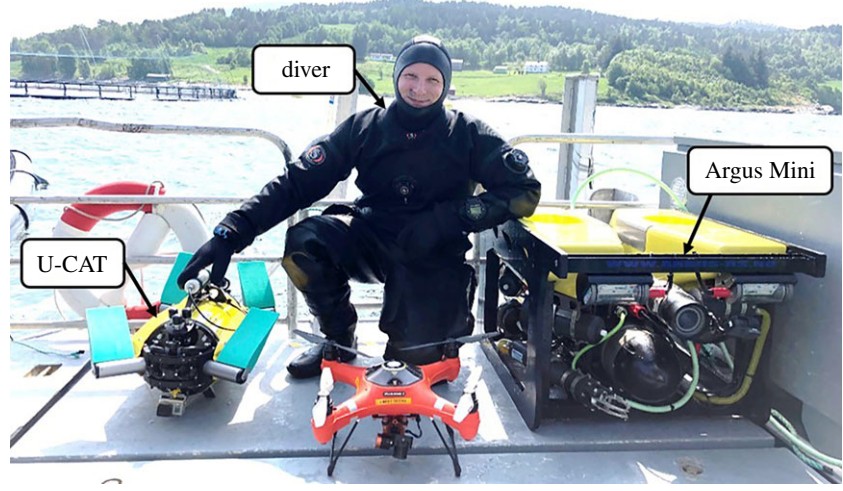

**Figure 1.** Mobile camera (GoPro3) platforms: the flipper-propelled U-CAT robot (left), human diver and the thruster-driven Argus Mini ROV (right).

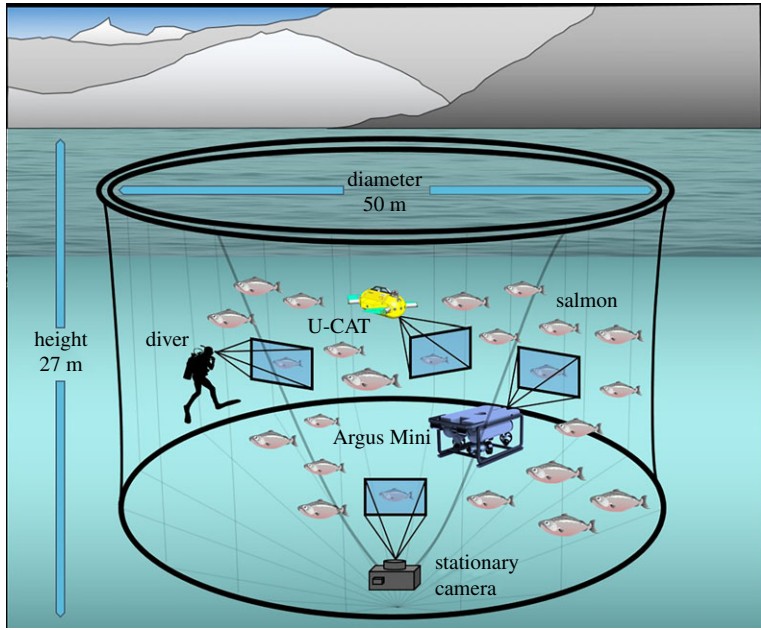

**Figure 2.** Illustration of the salmon farm sea cage test environment (not to scale). The stationary camera in the sea cage was used to analyse fish's tailbeat frequency and distance as the dependent parameter in the presence of the human diver, the flipper-propelled U-CAT and the thruster-driven Argus Mini. In addition, the tailbeat frequency was assessed using GoPro3 cameras mounted on the human diver as well as the U-CAT and Argus Mini underwater robots.

because the flippers generate a weaker wake than the large jet wakes produced by thrusters. As opposed to laboratory prototypes previously used in fish–robot interaction experiments, U-CAT is a field deployable robot. It has been tested in different natural conditions for prolonged periods of time and is capable of coping in underwater environments including currents and waves [30]. U-CAT is also smaller than most commercial inspection robots, and from the outcomes of [21], we also assume that a smaller size is beneficial for *in situ* observations. In this study, we compare the fish's reaction to U-CAT against standard technologies and practices in aquafarming using a stationary camera, a technical diver and Argus Mini, a commercial thruster-driven underwater robot (figure 1). Furthermore, we sought to answer if the design of U-CAT and its operation could be improved by investigating the fish's reaction in response to the locomotion mode and colour.

We analysed video footage from a stationary camera mounted in the net pen and from the GoPro3 cameras on the diver and the robots (figure 2). The camera footage allowed for the calculation of the two dependent variables assessed in this study: the tailbeat frequency of the fish and the distance

from the camera to the fish. Specifically, the footage from both the stationary bottom camera and mobile GoPro3 cameras mounted on the diver and the robots was used to calculate the fish tailbeat frequency, and the stationary camera footage was used for the distance.

# 2. Material and methods

## 2.1. Experimental setting

The experiments were conducted in a salmon farm at Korsneset, 110 km west of Trondheim, Norway (63° 8'27″ N, 8°13'49″ E), 28–31 May 2018. The study site was located in the outer areas of a typical Norwegian fjord environment with the main arms of the fjord, Vinjefjorden and Halsafjorden, stretching 40 km into the landmass to the east and south of the site, respectively. The fjord reaches depths down to about 500 m and is 2 km wide at the study site, while the depth below the sea cages ranges 40–120 m. The fjord location shelters the farm from severe ocean waves and currents, and the flow is mainly driven by the semi-diurnal tidal cycle which sustains typical differences in water level in the range 0.9–1.8 m. Previous site surveys have shown dominant eastward and westward currents in accordance with the general orientation of the fjord. Mean current speeds (10 min intervals) at the farm location were measured independently at $6.0 \pm 1.1 \, \text{cm s}^{-1}$ and $8.2 \pm 1.0 \, \text{cm s}^{-1}$, while maximum current speeds were $52 \, \text{cm s}^{-1}$ and $51 \, \text{cm s}^{-1}$, at 5 m and 25 m depth, respectively (K. Frank, SINTEF Ocean 2019, personal communication).

The farm consisted of 14 sea cages containing farmed Atlantic salmon (*Salmo salar*). The cages used in this study had an inner diameter of 50 m and a maximal depth of 27 m. The preparatory experiments where conducted in one sea cage and then moved to another for the rest of the test period because of better visibility. The weather conditions were good during the test period, sunny or partially cloudy, with mild wind.

The stock in the sea cage was measured a week before the experiments as part of the standard fish farm management procedure and was calculated to contain 188 000 individuals. Owing to the large number of fish, it is unlikely that the robot encountered the same individual twice, hence we consider the habituation bias to be negligible. The average weight of the fish (measured from 50 randomly collected individuals) was $330.5 \pm 109 \, \text{g}$. Assuming a healthy condition of a fish with Fulton's condition factor $k = 1.4$, the average length of fish was calculated as equal to $0.28 \pm 0.29 \, \text{m}$.

The sea cage also contained about 1000 cleaner fish (*Cyclopterus lumpus*). Their average length was 0.10 m and weight of 40 g, estimated from visual observations. The fish where automatically fed during the work day from a rotating feeding dispenser in the middle of the cage which distributed food over the surface. The site manager manually turned feeding on and off depending on his feedback from the stationary camera mounted in the middle of the net.

The sea cages were exposed to variable tides and currents through the fjord, which changed direction during the day and together with wind-induced wave action created a constantly changing local flow field. During the last two days of experiments, the wind changed direction and caused stronger waves. Therefore, visibility was considerably worse during the last day of the experiments. The test site was also exposed to occasional acoustic noise caused by pumps, feeding mechanisms, service vessels and maintenance equipment. Confounding factors which could not be controlled include naturally changing environmental conditions such as the weather and tides, as well as noise and vibrations caused by daily aquaculture operations.

A daily summary of the weather conditions was provided by the online service Yr, the joint service of the Norwegian Meteorological Institute and Norwegian Broadcasting Corporation [31]. Tides at the site location during the field trials were obtained via the Norwegian Mapping Authority [32]. A summary table of the environmental conditions is provided in table 1. The water temperature was 7°C at 10 m depth and fluctuated between 9°C and 11°C at 5 m depth over the duration of the field tests. The average May sea temperature at the test site is 8.4°C (minimum 6.3°C, maximum 10.6°C). During preliminary experiments, we observed that fish were unevenly distributed along the length and depth of the cage and tended to flock in a tight formation below 5 m depth, possibly because of the presence of a thermocline or halocline. Therefore, depth for all experiments was fixed at 5 m to encounter as many fish as possible for statistical analysis.

## 2.2. Description of underwater robots and cameras

U-CAT is an underwater vehicle actuated by four independent flippers. The vehicle's dimensions (including the fins) are $0.60 \times 0.57 \times 0.28 \, \text{m}$ and it weighs 18 kg. During all experiments, the robot was

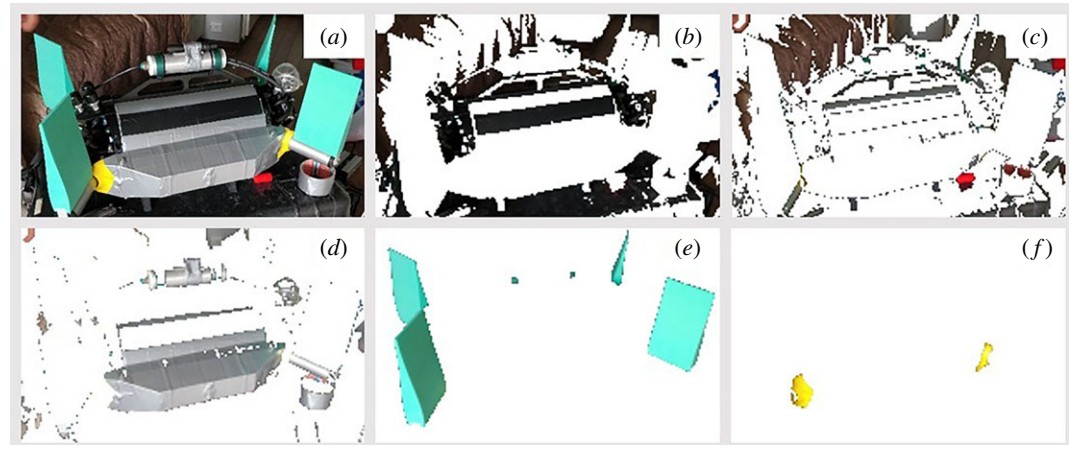

**Figure 3.** Example image taken of U-CAT with silver tape. (*a*) Original image, (*b*) colour class I, black background and surfaces not used in colour assessment, (*c*) colour class II, dark grey background and surfaces not used in colour assessment, (*d*) colour class III, duct tape and metallic surfaces used for U-CAT silver body colour assessment, (*e*) colour class IV, used for U-CAT flipper colour assessment, (*f*) colour class V, used for U-CAT yellow body colour assessment.

**Table 1.** Overview of environmental conditions at the sea cage study site. Weather conditions obtained from the station Kristiansund, 20 km west of the field study site. Tidal elevations are based on a datum taken from the mean sea level (1996–2014).

| month/ day | air temperature (°C) min/mean/max | wind speed (m/s) min/mean/max | humidity (%) min/mean/max | morning low tide (cm) | midday high tide (cm) | evening low tide (cm) | night high tide (cm) |
|---|---|---|---|---|---|---|---|
| 05/28 | 15.1/18.0/24.4 | 0.9/2.0/3.5 | 39/52.8/69.0 | −94 | 70 | −100 | 72 |
| 05/29 | 11.9/13.0/19.1 | 0.0/4.1/8.8 | 63.0/74.6/87.0 | −99 | 71 | −99 | 75 |
| 05/30 | 8.3/15.6/21.4 | 1.1/3.9/6.1 | 41.0/60.6/88.0 | −101 | 69 | −96 | 76 |
| 05/31 | 14.0/18.9/25.2 | 1.0/2.5/5.4 | 27.0/53.0/72.0 | −101 | 65 | −92 | 73 |

manually operated in tethered mode with the feedback from its built-in camera. The fish–robot interaction with U-CAT was investigated using two different locomotion modes: a hovering mode is used for high manoeuvrability, whereas the cruising mode allows for faster forward motion. The forward speed in hovering mode is 0.5 m s$^{-1}$ and in cruising mode 1 m s$^{-1}$. However, the sea cage was exposed to tidal currents and the speed of the vehicle varied depending on the direction of the flow. In one experiment, it was observed that it took twice as much time for the vehicle to drive the same route.

Argus Mini is a commercial, remotely operated robot powered by six thrusters. It has dimensions of $0.9 \times 0.65 \times 0.5$ m and weighs 90 kg.

Neither of the robots can directly control their relative speed over ground. Argus Mini can control the thruster load, which was manually controlled by the operator to keep it as constant as possible.

Both robots were equipped with an external GoPro3 camera. Footage from GoPro3 was used to measure fish tailbeat frequency in reaction to the intruder as they could provide better quality footage than the built-in cameras. The GoPro3 was attached to the robots to look forward and the diver was instructed to keep it possibly steadily looking forward.

The sea cage was equipped with a stationary feeding camera, adjusted approximately in the middle of the cage and connected to the central control deck of the fish farm (figure 2). The site manager of the farm could control the depth and view angle of the camera from the deck. When filming the robots above the camera, he could change the view angle to keep the robots and the diver in the camera field of view possibly long (figure 3).

## 2.3. Overview of experiments

The experimental plan is summarized in table 2 and was designed to make as many comparisons as possible given the constraints faced during field deployments. As an example, during experiments

**Table 2.** Comparison of experimental set-ups. The + sign indicates that the test under the specific conditions was performed. The (−) sign indicates that it was not realistic under those conditions (for example, it is not possible to traverse a transect with a passive robot). The (+) marks that the diver remained still along the transect.

| | diver (black) | Argus Mini (yellow) | U-CAT | | | | | | | |
| | | | passive | | hover | | cruise | | tow | |
| | | | yellow | silver | yellow | silver | yellow | silver | yellow | silver |
|---|---|---|---|---|---|---|---|---|---|---|
| ascending above stationary camera | + | + | + | + | + | + | − | − | − | − |
| transect traversing | (+) | + | − | − | + | + | + | + | + | + |

using U-CAT in cruising and towing mode, it is not possible to keep station and therefore recording from the stationary camera was not possible.

## 2.4. Stationary camera experiments

The Argus Mini and U-CAT underwater robots were driven to the middle of the cage and stopped once the stationary camera was visible from the robot's camera. The robots were then controlled to hover about 1 m above the camera for 5 min by the operator keeping the stationary camera at sight. Then the thrusters of the Argus Mini or the flippers of the U-CAT were switched off. As both robots are slightly positively buoyant, they slowly ascended to the surface. The diver was instructed to dive to the feeding camera, stay possibly motionless above the camera for 5 min and then ascent possibly slowly.

## 2.5. Robot and human diver dominant colour characterization

The impacts of the robot and human diver colour are considered as confounding factors in this study. However, it was possible to vary the colour of the main body of U-CAT by covering it with silver duct tape to assess if there was a significant change in the fish's response as compared to the yellow body colour. In order to characterize the colour of the robots and human diver, a simple method was applied which characterizes the colours using k-means clustering [33]. This method was chosen because it is suitable for field work where the ambient illumination conditions are highly variable and cannot be assessed throughout the investigation domain during the field study [34]. Images were taken in the field with a OnePlus A5010 smart phone's 16MP camera with f/1.7 aperture and 27 mm focal length lens set using automatic white balance adjustment and ISO values, which ranged from ISO 160 to 800.

First, a series of three different images were selected of the U-CAT, Argus Mini and human diver taken outdoors during the field study. Next, regions of interest (ROI) were created by selecting a bounding box around each of the three different objects. Finally, the ROI for U-CAT (yellow and silver separately), Argus Mini and the human diver were processed in an online image colour analysis programme (Image Color Summarizer, v. 078). The settings used were number of colour group clusters $k = 5$ and a precision of 200 pixels, which is the maximum allowable resolution of the resized image used in colour analysis. The choice of five clusters was made in order to provide two classes for the background (sea and terrain), one class for non-object surfaces (red connectors, railings, flooring, etc.) and two classes for the target object (U-CAT body and flippers). The summary results of the colour characterization are provided in table 3. Each of the object's colour classes is represented as the mean RGB, HSV and Laboratory colour spaces from the three ROIs, as well as $\Delta E$, which is the Euclidean measure of colour difference between the colours for each of the clusters in Laboratory space. The Argus Mini body ROIs produced two distinct classes associated with the body colour. This was due to the top surface being subject to direct illumination, with the sides being shaded. The colour classification results for Argus Mini therefore tended to have higher colour component standard deviations when compared to the other objects.

**Table 3.** Colour characterization of the human diver and underwater vehicles U-CAT and Argus Mini. Results are based on k-means clustering of colour groups from a series of three regions of interest cropped from field images obtained using a 16MP smart phone camera. The mean values of the colour groups are presented as three different colour spaces RGB, HSV and Laboratory, and the variability of each colour space is given as the Euclidean distance $\Delta E$, of the colour group in Laboratory colour space. The uncertainty of all colour values was estimated as the standard deviation of the individual colour components and is provided in parentheses.

| object | R | G | B | H | S | V | L | A | B | $\Delta E$ |
|---|---|---|---|---|---|---|---|---|---|---|
| diver | 18 (5.3) | 19 (3.7) | 19 (3.7) | 201 (4.5) | 16 (2.2) | 7 (1.6) | 5 (1.2) | −1 (0.5) | −1 (0.0) | 1.0 (0.7) |
| U-CAT yellow body | 234 (6.3) | 212 (11.5) | 57 (6.6) | 53 (0.9) | 76 (3.8) | 90 (6.6) | 84 (6.4) | −10 (1.9) | 74 (4.5) | 3.9 (1.4) |
| U-CAT silver body | 167 (9.9) | 168 (8.5) | 168 (8.0) | 201 (6.2) | 1 (0.8) | 66 (3.3) | 69 (2.9) | 0 (0.5) | −1 (0.5) | 0.7 (0.1) |
| U-CAT flippers | 78 (19.1) | 184 (18.7) | 163 (24.5) | 167 (5.6) | 59 (5.8) | 72 (7.1) | 68 (6.1) | −36 (1.4) | 2 (4.5) | 2.9 (0.8) |
| Argus mini yellow body | 220 (27.8) | 206 (31.7) | 92 (63.0) | 54 (1.6) | 58 (20.4) | 86 (9.9) | 82 (10.4) | −9 (2.3) | 54 (12.4) | 3.0 (0.6) |

## 2.6. Transect traversing

The robots were controlled to traverse a 20 m long transect in the sea cage, where all the experiments were recorded using a GoPro3 mounted on the robot as it moved. Before commencing each experiment, the robot descended to 5 m depth. Next, the operator navigated to the centre of the cage, keeping the net in the field of view. U-CAT was tested in passive, hover, cruise and towing modes. In towing mode, the flippers are switched off and the unactuated robot is towed manually. To create the silver colour, the U-CAT robot was covered with silver duct tape, including two black stripes (figure 3).

Instead of transect traversing, the diver was instructed to remain motionless on the transect because we observed during the experiments that a swimming diver would immediately repel the fish, and any resulting comparison would have had a trivial outcome. The transect experiments were performed with a minimum of two replicates, each in both directions.

## 2.7. Video analysis

In total, 350 GB of video (1505 min) was gathered over the course of all experiments. All videos were indexed, associated with the entries of the field protocol and pre-processed by removing frames where fish were not visible (mostly robot descending, ascending, hovering on the surface or near the net). As such, the total amount of videos was reduced to 50 GB (215 min) and the individual files were processed on a general-purpose computer. The GoPro3 video analysis was performed with the Kinovea® motion analysis software and the Loligo Systems® LoliTrack software was used for processing the stationary camera videos.

The GoPro3 camera videos from each experiment were processed using the Kinovea motion capture analysis tool. However, low visibility and the presence of many moving objects made it impossible to automatically track each individual fish. Owing to this, Kinovea was used to mark the fish as distinct objects and count time. The tailbeat frequency was determined manually, using the following procedure:

1. Each video was viewed beginning to end by the experimenter, and a video segment of interest was chosen subjectively to correspond to a series of video frames exhibiting the greatest abundance of fish.
2. Starting from the beginning of the video segment, the largest (closest to the camera) fish was marked, its length measured in pixels and the number of tailbeats were counted until the fish left the field of view.
3. The procedure was repeated for $n = 50$ individuals, and the produced .kva file with markers was used as input for further statistical analysis.

The analysis showed that both the tailbeat frequency and size of fish converged towards the mean value at around 25–30 individuals. A constant sample size of $n = 50$ individual fish was chosen for all analysis because it provided a sample size satisfying a desired precision of ±10% of the dependent variable mean (tailbeat frequency or distance) across all experiments, 95% of the time. This level of precision was chosen because it is consistent with recent advances in automated image analysis algorithms to estimate fish tailbeat frequency [35].

Repeatability analysis was performed separately for the U-CAT and Argus Mini underwater robots to confirm that there was no subjective bias in video processing. Videos from two repetitions of the same conditions where processed by the same experimenter by calculating the tailbeat frequency. Two datasets from replicated runs ($n = 29$, each run) of the Argus Mini experiments were analysed with one-way analysis of variance (ANOVA, $p = 0.99$, $F < 0.001$). In addition, two replicated runs ($n = 50$, each run) using the yellow U-CAT robot in hovering mode were analysed with one-way ANOVA ($p = 0.88$, $F = 0.02$). In both statistical analyses, the dependent variable was the mean tailbeat frequency, and the type of underwater robot was the independent variable. Based on these results (figure 4), we concluded that the tailbeat frequency data provided reproducible, fish–robot interaction metrics in the tested salmon aquaculture pen environment. The significance level for all ANOVA tests was set at $p \leq 0.05$, following a comprehensive video-based laboratory study fish–robot interactions [14]. *Post hoc* tests were used on results where a significant main effect of an independent variable (e.g. robot versus human diver, robot colour, actuation mode) was found.

Underwater video obtained from the vertically oriented stationary camera was used to determine the fish tailbeat frequency (TB s$^{-1}$) as well as the straight-line distance from the robots and the diver. Fish on the same horizontal level as the robot were defined comparing the sizes of fish and the robots (both known variables). All fish located along the same horizontal level as the robot (e.g. at the same distance from the camera) were marked using threshold colours by the LoliTrack software. Fish

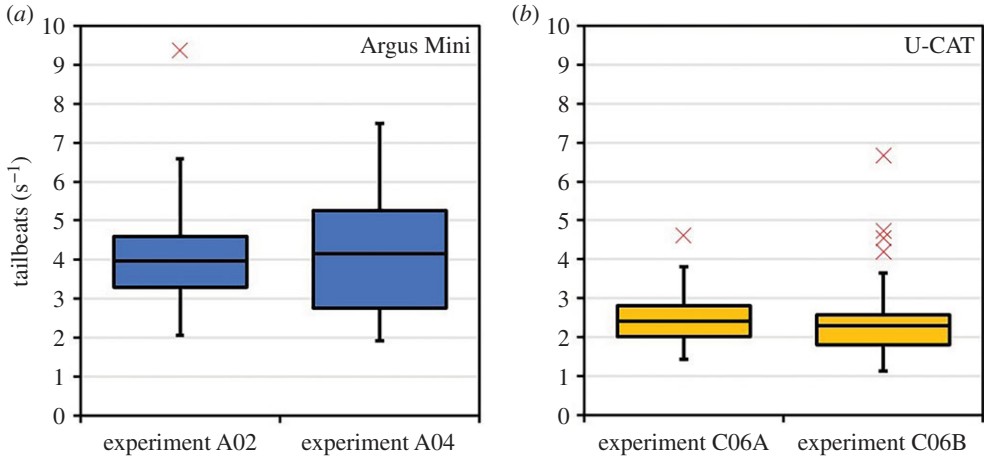

**Figure 4.** Results of the repeatability analysis as box and whisker plots. Boxes represent the interquartile range (IQR) over the 25th to 75th percentiles, the centreline corresponds to the median, error bars extend from the IQR up to a factor of 1.5 from the IQR and outliers are shown in red. (*a*) Replicated Argus Mini experiments as box and whisker plots. (*b*) Plots of the U-CAT experimental replicates.

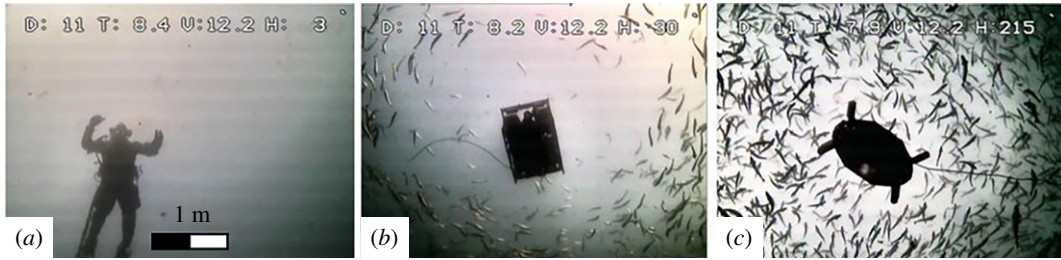

**Figure 5.** Frames from stationary camera filming the diver (*a*), Argus Mini (*b*) and U-CAT (*c*).

distance from the robots was measured using the Universal Desktop Ruler software. Tailbeats of fish were counted manually within the LoliTrack player along 3–4 m trajectories.

## 3. Results

All stationary camera videos were analysed using one-way ANOVA to compare the reaction of $n = 50$ individual fish in the presence of U-CAT, Argus Mini or the human diver (figure 5). The reference video was therefore obtained with no intruder present.

The analysis of the tailbeat frequency is presented in figure 6 and reveals that there is a significant difference in tailbeat frequency in the presence of U-CAT and other intruders (Argus Mini and the diver). One-way ANOVA was used to assess variations between the fish mean tailbeat frequencies (dependent variable) in response to the intruder (independent variable). It was observed that the mean tailbeat frequencies were the slowest in the vicinity of the U-CAT robot, and even slower when no object was present ($p \leq 0.05$, $F = 94.77$, all scenarios).

One-way ANOVA was also performed to assess variations in the response to actuation. Specifically, the mean tailbeat frequency was taken as the dependent variable, and the robot type and actuation mode was evaluated as the independent variable. It was observed that for the majority of experiments, fish did not react strongly to changes in the actuation. There was no significant difference in the response to Argus Mini while actively holding station or floating passively ($p = 0.22$, $F = 1.51$). However, the difference between the fish's reaction to U-CAT with its flippers moving and a passively floating U-CAT was small, but significant ($p \leq 0.05$, $F = 9.29$).

Next, we carried out an additional analysis by calculating the straight-line distance of fish from the underwater robots. Similar to the tailbeat frequency, a one-way ANOVA ($n = 50$) was used to study the variations in response to actuation or passive conditions, as shown in figure 7. The dependent variable was the mean straight-line distance from robot to fish, and the independent variable was the underwater robot type and its actuation mode. On average, fish remained some 0.5–0.7 m from U-CAT, corresponding to 1.5–2.2 body lengths (BL), whereas the distance from Argus Mini was on average

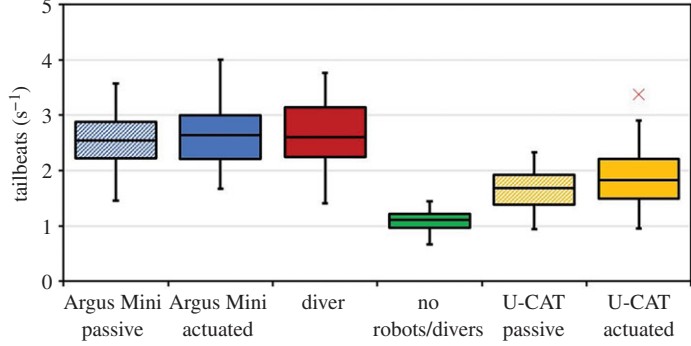

**Figure 6.** Box and whisker plot of the fish swimming response (mean tailbeats s$^{-1}$) in the presence of robots and divers compared to fish swimming response when no object is present ($n = 50$, all scenarios). Boxes represent the interquartile range (IQR) over the 25th to 75th percentiles, the centreline corresponds to the median, error bars extend from the IQR to a factor of 1.5 the IQR and outliers are shown in red.

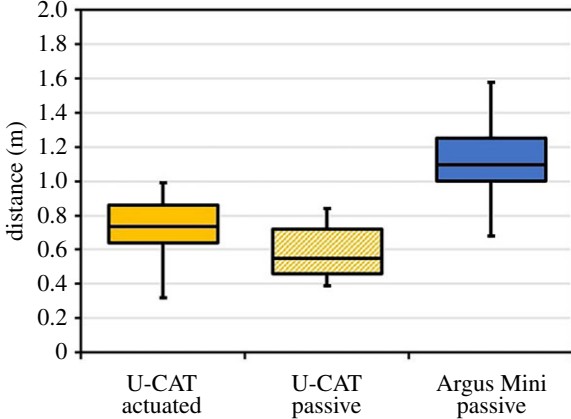

**Figure 7.** Box and whisker plot of the mean straight-line distance between the fish and the robot, in metres. Boxes represent the interquartile range (IQR) over the 25th to 75th percentiles, the centreline corresponds to the median, error bars extend from the IQR to a factor of 1.5 the IQR, no outliers were present in these datasets. Multiple comparison of the mean distances using the Tukey–Kramer *post hoc* test found that all means were significantly different ($p \leq 0.05$, all cases).

1.09 m, nearly four BL. The distance from the hovering U-CAT was somewhat larger than from the passively floating U-CAT. In all cases, the fish reactions were found to be significantly different from each other ($p \leq 0.05$, $F = 144.65$).

The camera mounted on the diver, Argus Mini and the U-CAT allows for the analysis of fish reactions to a moving platform, example imagery from the GoPro3 cameras is provided in figure 8. The tailbeat frequencies for interaction with the human diver, Argus Mini and yellow-coloured U-CAT in hover mode were evaluated using one-way ANOVA ($n = 50$) to assess variations between the mean fish response (dependent variable: tailbeat frequency) and intruder types (independent variable: U-CAT, Argus Mini or human diver). The distribution of the mean tailbeat frequencies for each type of intruder are illustrated in figure 9, where a significant difference was detected ($p \leq 0.05$, $F = 22.89$). *Post hoc* analyses using the Tukey–Kramer test showed that the mean tailbeat frequency of the yellow U-CAT intruder type was significantly different from the mean tailbeat frequencies observed for the Argus Mini and human diver types ($p \leq 0.05$, both cases).

It was found that fish had the lowest mean tailbeat frequency in the presence of U-CAT and the highest in the presence of the Argus Mini. Considering outliers as observations outside of a factor of 1.5 from the IQR, it was also found that there was an increase in the number of statistical outliers when compared to the tailbeat frequency data obtained using only the stationary camera. Upon reviewing the individual outlier instances on the original video recordings, it was determined that they correspond to specific occasions where fish are rapidly fleeing. The number of outlier instances caused by fleeing fish was found to be highest in the presence of the diver (five outliers) and remained low for the actuated U-CAT (one outlier) and Argus Mini (one outlier).

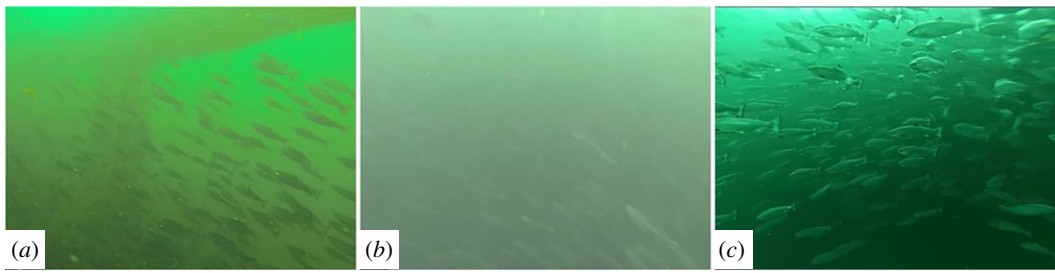

**Figure 8.** Frames from on board GoPro3 camera mounted on the diver (*a*), Argus Mini (*b*) and U-CAT (*c*).

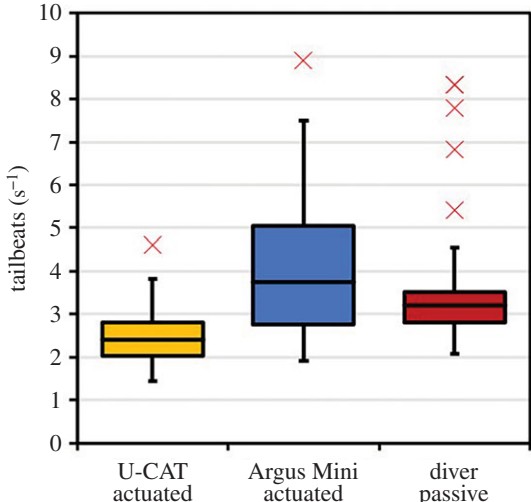

**Figure 9.** Box and whisker plots of the observed tailbeat frequencies from the GoPro3 cameras of the actuated U-CAT and Argus Mini underwater robots, as well as the motionless human diver. Boxes represent the interquartile range (IQR) over the 25th to 75th percentiles, the centreline corresponds to the median, error bars extend from the IQR to a factor of 1.5 the IQR and outliers are shown marked in red.

## 3.1. Fish tailbeat frequency in reaction to U-CAT locomotion modes

The tailbeat frequency of fish in the presence of the U-CAT underwater robot was tested in three locomotion modes: slow hover, fast cruise and towing mode. Statistical analyses of the variability between actuation modes were then compared to a passive U-CAT using a one-way ANOVA when the robot was coloured yellow ($p \leq 0.05$, $F = 51.25$) as well as covered with silver tape ($p \leq 0.05$, $F = 4.7$). The mean tailbeat frequencies for each of the three actuation modes were taken as the dependent variables, and the independent variable was the robot colour. *Post hoc* analyses using the Tukey–Kramer test showed that for a yellow U-CAT, the mean tailbeat frequency during cruise mode was significantly different from the hovering and towing actuation modes ($p \leq 0.05$). However, the Tukey–Kramer *post hoc* test applied to the silver U-CAT showed no significant differences between the mean tailbeat frequencies during cruising as compared to hovering ($p = 0.37$) or towing ($p = 0.20$). The distributions of both yellow and silver U-CAT experiments are illustrated in figure 10. It is also worth noting that the experimental runs using the silver-coloured U-CAT resulted in a larger number of outliers, the majority in the towing mode. Reviewing those instances on recordings, they corresponded to occasions where fish were initially close to the robot and then fled rapidly as it approached.

## 4. Discussion

The general conclusion from this study is that the experimental robot U-CAT permits observing fish in an aquafarm at close distance (0.5–1.5 m), while conventional observation methods from moving platforms are significantly more disturbing or do not permit observations at all. However, the presence of U-CAT still invokes a behavioural response from the fish. The study did not find conclusive evidence that the

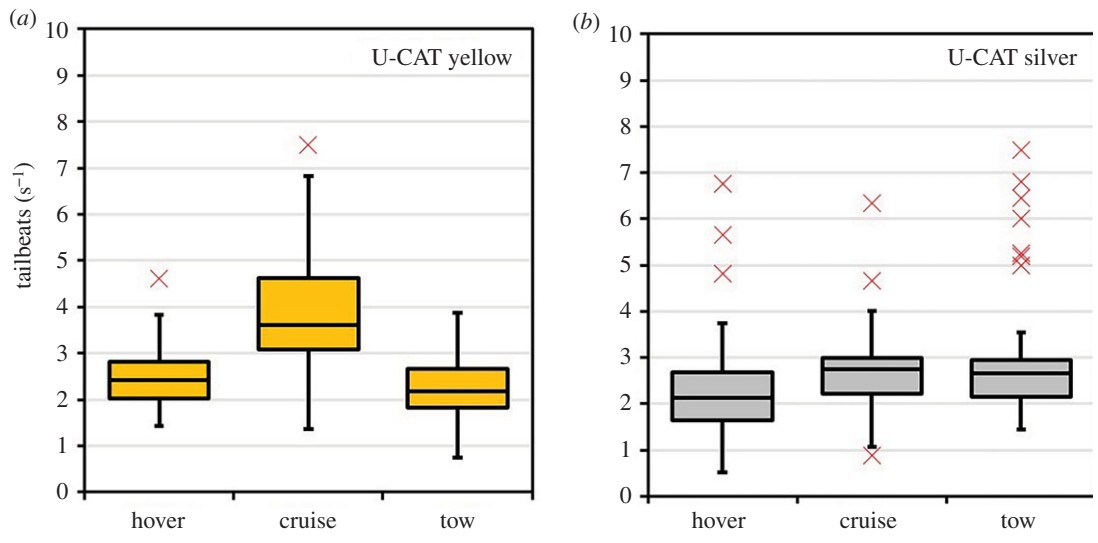

**Figure 10.** Comparative box and whisker plots of the fish response to U-CAT considering different operating modes and yellow and silver colours, from GoPro3 camera video analysis. The boxes represent the interquartile range (IQR) over the 25th to 75th percentiles, the centreline corresponds to the median, error bars extend beyond a factor of 1.5 the IQR and outliers are shown marked in red.

locomotion mode, colour, sound or speed had a large effect on animal behaviour. This finding may be used in future studies to guide the use of underwater robots in fish monitoring activities.

The avoidance reaction of fish (measured both in TB s$^{-1}$ and straight-line distance) is strongest in the presence of the diver. It is hard to say which stimuli from the diver caused most stress (size, shape, colour and bubbles from the breather). However, it can be concluded that the standard procedure of fish cage inspection by commercial divers is highly disturbing for the fish and by diving it is not possible to observe fish in their natural state. This conclusion is supported by other studies conducted in the wild with different fish species [36–38].

The presence of Argus Mini (a commercial underwater robot used for cage inspection) caused significantly larger avoidance reaction than U-CAT. Our initial hypothesis was that the flipper-based design of the vehicle is less disturbing to fish than propellers. However, fish tended to avoid the passive Argus Mini more so than an actuated U-CAT so it is also entirely possible that the behavioural difference was more or mostly caused by the size and/or shape but not the locomotion principle.

At the same time, we did not find a significantly different reaction to U-CAT in different locomotion modes and colours unless the yellow U-CAT underwater robot was moving in a cruising mode. Cruising mode of the robot is faster so it may indicate that fish are avoiding more a faster moving object of bright colour. Regardless of the colour, fish seem to be most relaxed in the vicinity of the robot in hover mode (which is slower than cruising mode). It is noticeable though that the silver robot in towing mode has a large number of statistical outliers (fish rapidly accelerating). This may mean that the fish have difficulty observing a silver and quiet robot coming, and attempt to escape from it only when it is very close. That the robot in different locomotion modes did not significantly affect fish behaviour suggests that fish in this study are indifferent to the hydrodynamic disturbance from flippers and also that the exact locomotion principle (biomimetic or not) is not a decisive factor. Neither did fish react significantly differently to propeller motion of Argus Mini which may suggest that propeller-driven vehicles are still a simple, affordable and reliable option in animal monitoring.

Since in towing mode, the motors are switched off, we can conclude that the fish are not sensitive to the noise from the U-CAT's DC motors actuating the fins. One possible explanation is that fish do not respond to the audio stimuli with the particular characteristics from the DC motors. Audiograms from wild and adult salmon typically range from 100 to 800 Hz [39]. If the motor noise is largely outside of this range, then it is very likely that the fish cannot hear it. The other possible explanation is that farmed fish are used to high ambient noise level. Sound propagates well underwater and there is a significant ambient noise in the farm with varying character from pumps, ships, etc. The third possible explanation is reported in [40] and suggests that farmed salmon might have developed hearing impairment in the course of breeding.

In general, we did not find conclusive support to our initial hypothesis that the flipper-based locomotion disturbs fish less and is therefore more suitable for fish observations and therefore

conclude that the hydrodynamic or visual cues from different locomotion patterns and colours are not crucial design parameters. It can equally be that the decisive factors altering fish behaviour are more rudimentary cues such as size and speed of the vehicle. The field test with a biomimetic robot in a wild-caught mackerel school reached a similar conclusion [21]. We find it important to point out that fish have no idea whether the robot is supposed to be biomimetic or not and might react to different cues than those that appear salient for humans. As [7] also emphasizes, many robot–animal interaction experiments are successfully conducted also with non-biomimetic replicas. This aspect is also critically addressed in [41] in the context of the exploitation of trivial or naive ideas of similarity. Some cues (such as shape and colour) might be important for humans (who mostly rely on visual cues) whereas animals might react to different stimuli.

These results have direct implications for practical applications. Some features of biomimetic underwater robots are complicated and include undulating actuators, high manoeuvrability with many control surfaces, which can drive up the cost and delay the exploitation of the technology in the field. Considering the current application, there is a clear demand from industry for more automation, and it is of practical importance to find the simplest way of answering it.

Data accessibility. The videos selected used for analysis in paper and the associated .kva files from Kinovea are indexed as https://doi.org/10.5061/dryad.x0k6djhfs [42].

Authors' contributions. M.K. conceived the concept and defined the research problem, planned and conducted the experiments, performed GoPro3 image analysis, interpreted the results and wrote the paper. R.G. contributed to planning the experiments, performed the experiments and wrote the paper. J.A.T. contributed to planning the experiments, statistical analysis, interpretation of the results and writing the paper. A.T. chose methods for behaviour analysis, performed video analysis from the stationary camera and interpreted the results. J.A.A. conceived the concept and defined the research problem, participated in experiments and wrote the paper.

Competing interests. We declare we have no competing interests.

Funding. European Commission through H2020 Research Infrastructures grant no. 65283 AQUAEXCEL2020 TNA research infrastructure project using the SINTEF ACE facilities of SINTEF Ocean AS operated by SalMar AS in Korsneset, Norway. Research Council of Norway through the Centres of Excellence funding scheme, project number 223254 - NTNU AMOS and the Estonian Research Council grant nos. IUT-339 and PUT-1690.

Acknowledgements. The experiments conducted in this paper are approved by the Ethics Committee of AQUAEXCEL2020 TNA.

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
