## [Reviewer comments · Royal Society Open Science]

Review History

RSOS-191220.R0 (Original submission)

Review form: Reviewer 1

Is the manuscript scientifically sound in its present form?

No

Are the interpretations and conclusions justified by the results?

Yes

Is the language acceptable?

Yes

Do you have any ethical concerns with this paper?

No

Have you any concerns about statistical analyses in this paper?

Yes

Recommendation?

Major revision is needed (please make suggestions in comments)

Comments to the Author(s)

The paper “Underwater Robot Interaction with a Large Fish Aggregation” by Kruusmaa et al. investigates fish reaction to underwater robots in a large aggregation of fish with a purpose to guide the design of underwater robots for fish observations.

Although I found this article rather interesting, especially for its application in the real world, there are important issues that must be addressed by authors in order to have a paper suitable for publication.

Also relevant literature has not been reported.

Pag.3 line 2: Please include these recent reviews on animal-robot interactions:

Romano, D., Donati, E., Benelli, G., & Stefanini, C. (2019). A review on animal-robot interaction: from bio-hybrid organisms to mixed societies. *Biological cybernetics*, 113(3), 201-225.

Porfiri, M. (2018). Inferring causal relationships in zebrafish-robot interactions through transfer entropy: A small lure to catch a big fish. *Animal Behavior and Cognition*, 5(4), 341-367.

Pag.3 line 11: Also, robots were used in predator-prey interactions, as well as to encode desired behaviour in insects of economic importance. Please include these relevant works in your article: Cianca, V., Bartolini, T., Porfiri, M., & Macrì, S. (2013). A robotics-based behavioral paradigm to measure anxiety-related responses in zebrafish. *PLoS One*, 8(7), e69661.

Ladu, F., Bartolini, T., Panitz, S. G., Chiarotti, F., Butail, S., Macrì, S., & Porfiri, M. (2015). Live predators, robots, and computer-animated images elicit differential avoidance responses in zebrafish. *Zebrafish*, 12(3), 205-214.

El Khoury, R., Ventura, R. B., Cord-Cruz, G., Ruberto, T., & Porfiri, M. (2018, March). Interactive experiments in a robotics-based platform to simulate zebrafish response to a predator. In *Bioinspiration, Biomimetics, and Bioreplication VIII* (Vol. 10593, p. 1059301). International Society for Optics and Photonics.

Romano, D., Benelli, G., & Stefanini, C. (2017). Escape and surveillance asymmetries in locusts exposed to a Guinea fowl-mimicking robot predator. *Scientific Reports*, 7(1), 12825.

Romano, D., Benelli, G., & Stefanini, C. (2019). Encoding lateralization of jump kinematics and eye use in a locust via bio-robotic artifacts. *Journal of Experimental Biology*, 222(2), jeb187427.

Pag.3 line 16: closed-loop control were used to enhance the degree of biomimicry of a robotic fish to interact with zebrafish:

Kim, C., Ruberto, T., Phamduy, P., & Porfiri, M. (2018). Closed-loop control of zebrafish behaviour in three dimensions using a robotic stimulus. *Scientific reports*, 8(1), 657.

Pag.3 line 20: Robots were used to investigate and modulate the aggressive and courthship behaviours of a highly territorial species such as Siamese fighting fish to multimodals cues including light, an inedited signal in this fish:

Romano, D., Benelli, G., Donati, E., Remorini, D., Canale, A., & Stefanini, C. (2017). Multiple cues produced by a robotic fish modulate aggressive behaviour in Siamese fighting fishes. *Scientific reports*, 7(1), 4667.

Romano, D., Benelli, G., Hwang, J. S., & Stefanini, C. (2019). Fighting fish love robots: mate discrimination in males of a highly territorial fish by using female-mimicking robotic cues. *Hydrobiologia*, 833(1), 185-196.

Pag.3 line 36: Please, tone down this statement. You are carrying out an experiment where you use a biometric artifcat to observe in the field fish without disturbing them. This is far from a real ecological interaction(e.g. social, courtship, agonistic, predatation, parasitic, etc.). In addition, please clarify that robots you are using are not biomimetic, so they are not intended at to be perceived as a conspecific in the group of animals but it has just bioinspired features mainly focused on the robots locomotion.

Pag.3 line 37-pag.4 line 5: Please, reduce this part describing what is a fish farm and what are its potentials. It decreases the readability of the manuscript.

Pag. 4 lines 23-28: Put this part in "Materials and methods" section.

Pag.5 line 49-50: Please, provide more rigorous and detailed information on the experimental conditions.

what was the average velocity of water current?

what time of the day was selected to carry out tests?

If you have some information about light conditions would be greatly beneficial for your study.

how long did the test last?

How many tests were performed?

Pag. 6 line 28: Inspired to which animal? This is important to report in an animal-robot experiment. Also, is important to clarify that your robot is not reproducing features of the animal model studied.

Pag. 6 lines 45-49: Please, provide color measurements for each robot. This could importantly affect the response of fish. Why was the color different for each tested robot? How can you dissect the effect of color (vision is one of the most important sense in fish), of different robots from the effect produced by other features considered?

Pag. 9 line 11: How was the distribution of your data? Maybe a non-parametric approach would be more appropriate.

Furthermore, a description of the experiment with animals has not been provided and just vague descriptions have been included.

A careful English revision is needed.

I hope my suggestions can help authors to improve their scientifically sounding work.

Review form: Reviewer 2

Is the manuscript scientifically sound in its present form?

No

Are the interpretations and conclusions justified by the results?

Yes

Is the language acceptable?

No

Do you have any ethical concerns with this paper?

No

Have you any concerns about statistical analyses in this paper?

Yes

Recommendation?

Major revision is needed (please make suggestions in comments)

Comments to the Author(s)

The manuscript „Underwater Robot Interaction with a Large Fish

Aggregation" by Kruusmaa investigated live fish responses in an aquaculture facility towards moveable objects like divers and robots. As pointed out by the authors, any robot-fish interactions are investigated in the lab and more field-based analysis on the possible application of robots are needed. The results of the current paper show that a smaller, flipper-propelled robot (U-CAT) elicits less avoidance reactions of the fish compared to a human diver and another, bigger thruster-propelled robot. This is a very interesting study and the acquired knowledge will for sure foster new developments in the field of bioinspired robots as well as robot-animal interaction. However, I feel that the manuscript cannot be accepted for publication in its current form as the statistical analyses are not presented at a sufficiently high scientific level. For example, only p-values are provided with no information whether they stem from the global ANOVA models or subsequent posthoc tests. No F-Values or other properties of the ANOVA results are shown. This makes it hard to judge whether the effects are really there or not. Furthermore, I could not find a clear statement on the sample size except an N=50 for all ANOVAS and a 50 GB video file size. What does N=50 mean in the respective cases? Number of analyzed video snapshots or trial runs? How was the sample size distributed among treatments? equally? Also, figure legend are way to minimalistic to full understand what is shown (Median or mean? What do the box and whiskers mean?). I strongly recommend putting more details into the description of the statistics!

Specific comments:

P2L46: Sentence "At the same time..." is not needed and can be deleted

P3L14: Golden Shiners

P3L50 food instead of feed?

P3L57: Better connection from environmental variation to robots need.

P6L40: Please be consistent: either report length as cm or m

P7L12: Table legend: More details are needed to understand that table. What do plus and minus mean?

P8L35: 350 GB: Minutes of observation would be a much better proxy for the amount of raw data acquired as 4K videos often have 2 GB per minute or even more but this highly depends on the camera model used.

P9L1: What does that mean "converged towards the mean value at around 25-30 fish encounters?" In addition, N=50, is this the number of experimental runs in total (when considering one run as a 5 min sequence that was described before above) or is it 50 analyzed pics? per treatment? in total? Please clarify.

P9L8: Repeatability analysis can be put into the supplement.

P10L37: Please provide statistical evidence (model values like F-values, mean squares, etc).

P10L44: P=0.04: The authors mention a small difference but also a significantly small P-value. Please clarify.

P11L21: Figure 4 legend: Please tell the sample size of analyzed frames and test runs per treatment. Tell what the box and the whiskers mean.

P11L26: "Analysis of the distance..." This sentence is incorrect. Fig. 5 does not show any analysis. It shows data!

P12L30: The description of the ANOVA results reads strange. Please revise.

P12L8: P-values smaller than 0.001 are reported as P<0.001.

Decision letter (RSOS-191220.R0)

08-Sep-2019

Dear Dr Kruusmaa,

The editors assigned to your paper ("Underwater Robot Interaction with a Large Fish Aggregation") have now received comments from reviewers. We would like you to revise your

paper in accordance with the referee and Associate Editor suggestions which can be found below (not including confidential reports to the Editor). Please note this decision does not guarantee eventual acceptance.

Please submit a copy of your revised paper before 01-Oct-2019. Please note that the revision deadline will expire at 00.00am on this date. If we do not hear from you within this time then it will be assumed that the paper has been withdrawn. In exceptional circumstances, extensions may be possible if agreed with the Editorial Office in advance. We do not allow multiple rounds of revision so we urge you to make every effort to fully address all of the comments at this stage. If deemed necessary by the Editors, your manuscript will be sent back to one or more of the original reviewers for assessment. If the original reviewers are not available, we may invite new reviewers.

- Data accessibility

<http://datadryad.org/submit?journalID=RSOS&manu=RSOS-191220>

- Competing interests

- Authors' contributions

All submissions, other than those with a single author, must include an Authors' Contributions section which individually lists the specific contribution of each author. The list of Authors

should meet all of the following criteria; 1) substantial contributions to conception and design, or acquisition of data, or analysis and interpretation of data; 2) drafting the article or revising it critically for important intellectual content; and 3) final approval of the version to be published.

- Acknowledgements

- Funding statement

on behalf of Dr Manoj Srinivasan (Associate Editor) and R. Kerry Rowe (Subject Editor)
openscience@royalsociety.org

Associate Editor's comments (Dr Manoj Srinivasan):

Associate Editor: 1

Comments to the Author:

The reviewers request some additional information regarding the study to make the article more complete. Additional discussion of prior lab-based work may also be indicated - one reviewer suggests additional references, some of which you may consider if they are relevant.

Comments to Author:

Reviewers' Comments to Author:

Reviewer: 1

Comments to the Author(s)

The paper "Underwater Robot Interaction with a Large Fish Aggregation" by Kruusmaa et al. investigates fish reaction to underwater robots in a large aggregation of fish with a purpose to guide the design of underwater robots for fish observations.

Although I found this article rather interesting, especially for its application in the real world, there are important issues that must be addressed by authors in order to have a paper suitable for publication.

Also relevant literature has not been reported.

Pag.3 line 2: Please include these recent reviews on animal-robot interactions:

Romano, D., Donati, E., Benelli, G., & Stefanini, C. (2019). A review on animal-robot interaction: from bio-hybrid organisms to mixed societies. *Biological cybernetics*, 113(3), 201-225.

Porfiri, M. (2018). Inferring causal relationships in zebrafish-robot interactions through transfer entropy: A small lure to catch a big fish. *Animal Behavior and Cognition*, 5(4), 341-367.

Pag.3 line 11: Also, robots were used in predator-prey interactions, as well as to encode desired behaviour in insects of economic importance. Please include these relevant works in your article: Cianca, V., Bartolini, T., Porfiri, M., & Macrì, S. (2013). A robotics-based behavioral paradigm to measure anxiety-related responses in zebrafish. *PLoS One*, 8(7), e69661.

Ladu, F., Bartolini, T., Panitz, S. G., Chiarotti, F., Butail, S., Macrì, S., & Porfiri, M. (2015). Live predators, robots, and computer-animated images elicit differential avoidance responses in zebrafish. *Zebrafish*, 12(3), 205-214.

El Khoury, R., Ventura, R. B., Cord-Cruz, G., Ruberto, T., & Porfiri, M. (2018, March). Interactive experiments in a robotics-based platform to simulate zebrafish response to a predator. In *Bioinspiration, Biomimetics, and Bioreplication VIII* (Vol. 10593, p. 1059301). International Society for Optics and Photonics.

Romano, D., Benelli, G., & Stefanini, C. (2017). Escape and surveillance asymmetries in locusts exposed to a Guinea fowl-mimicking robot predator. *Scientific Reports*, 7(1), 12825.

Romano, D., Benelli, G., & Stefanini, C. (2019). Encoding lateralization of jump kinematics and eye use in a locust via bio-robotic artifacts. *Journal of Experimental Biology*, 222(2), jeb187427.

Pag.3 line 16: closed-loop control were used to enhance the degree of biomimicry of a robotic fish to interact with zebrafish:

Kim, C., Ruberto, T., Phamduy, P., & Porfiri, M. (2018). Closed-loop control of zebrafish behaviour in three dimensions using a robotic stimulus. *Scientific reports*, 8(1), 657.

Pag.3 line 20: Robots were used to investigate and modulate the aggressive and courthship behaviours of a highly territorial species such as Siamese fighting fish to multimodals cues including light, an inedited signal in this fish:

Romano, D., Benelli, G., Donati, E., Remorini, D., Canale, A., & Stefanini, C. (2017). Multiple cues produced by a robotic fish modulate aggressive behaviour in Siamese fighting fishes. *Scientific reports*, 7(1), 4667.

Romano, D., Benelli, G., Hwang, J. S., & Stefanini, C. (2019). Fighting fish love robots: mate discrimination in males of a highly territorial fish by using female-mimicking robotic cues. *Hydrobiologia*, 833(1), 185-196.

Pag.3 line 36: Please, tone down this statement. You are carrying out an experiment where you use a biometric artifcat to observe in the field fish without disturbing them. This is far from a real ecological interaction(e.g. social, courtship, agonistic, predatation, parasitic, etc.). In addition, please clarify that robots you are using are not biomimetic, so they are not intended at to be perceived as a conspecific in the group of animals but it has just bioinspired features mainly focused on the robots locomotion.

Pag.3 line 37-pag.4 line 5: Please, reduce this part describing what is a fish farm and what are its potentials. It decreases the readability of the manuscript.

Pag. 4 lines 23-28: Put this part in "Materials and methods" section.

Pag.5 line 49-50: Please, provide more rigorous and detailed information on the experimental conditions.

what was the average velocity of water current?

what time of the day was selected to carry out tests?

If you have some information about light conditions would be greatly beneficial for your study.

how long did the test last?

How many tests were performed?

Pag. 6 line 28: Inspired to which animal? This is important to report in an animal-robot experiment. Also, is important to clarify that your robot is not reproducing features of the animal model studied.

Pag. 6 lines 45-49: Please, provide color measurements for each robot. This could importantly affect the response of fish. Why was the color different for each tested robot? How can you dissect the effect of color (vision is one of the most important sense in fish), of different robots from the effect produced by other features considered?

Pag. 9 line 11: How was the distribution of your data? Maybe a non-parametric approach would be more appropriate.

Furthermore, a description of the experiment with animals has not been provided and just vague descriptions have been included.

A careful English revision is needed.

I hope my suggestions can help authors to improve their scientifically sounding work.

Reviewer: 2

Comments to the Author(s)

The manuscript „Underwater Robot Interaction with a Large Fish Aggregation“ by Kruusmaa investigated live fish responses in an aquaculture facility towards moveable objects like divers and robots. As pointed out by the authors, any robot-fish interactions are investigated in the lab and more field-based analysis on the possible application of robots are needed. The results of the current paper show that a smaller, flipper-propelled robot (U-CAT) elicits less avoidance reactions of the fish compared to a human diver and another, bigger thruster-propelled robot. This is a very interesting study and the acquired knowledge will for sure foster new developments in the field of bioinspired robots as well as robot-animal interaction. However, I feel that the manuscript cannot be accepted for publication in its current form as the statistical analyses are not presented at a sufficiently high scientific level. For example, only p-values are provided with no information whether they stem from the global ANOVA models or subsequent posthoc tests. No F-Values or other properties of the ANOVA results are shown. This makes it hard to judge whether the effects are really there or not. Furthermore, I could not find a clear statement on the sample size except an N=50 for all ANOVAS and a 50 GB video file size. What does N=50 mean in the respective cases? Number of analyzed video snapshots or trial runs? How was the sample size distributed among treatments? equally? Also, figure legend are way to minimalistic to full understand what is shown (Median or mean? What do the box and whiskers mean?). I strongly recommend putting more details into the description of the statistics!

Specific comments:

P2L46: Sentence “At the same time...” is not needed and can be deleted

P3L14: Golden Shiners

P3L50 food instead of feed?

P3L57: Better connection from environmental variation to robots need.

P6L40: Please be consistent: either report length as cm or m

P7L12: Table legend: More details are needed to understand that table. What do plus and minus mean?

P8L35: 350 GB: Minutes of observation would be a much better proxy for the amount of raw data acquired as 4K videos often have 2 GB per minute or even more but this highly depends on the camera model used.

P9L1: What does that mean "converged towards the mean value at around 25-30 fish encounters?" In addition, N=50, is this the number of experimental runs in total (when considering one run as

a 5 min sequence that was described before above) or is it 50 analyzed pics? per treatment? in total? Please clarify.

P9L8: Repeatability analysis can be put into the supplement.

P10L37: Please provide statistical evidence (model values like F-values, mean squares, etc).

P10L44: $P=0.04$: The authors mention a small difference but also a significantly small P-value. Please clarify.

P11L21: Figure 4 legend: Please tell the sample size of analyzed frames and test runs per treatment. Tell what the box and the whiskers mean.

P11L26: "Analysis of the distance..." This sentence is incorrect. Fig. 5 does not show any analysis. It shows data!

P12L30: The description of the ANOVA results reads strange. Please revise.

P12L8: P-values smaller than 0.001 are reported as $P<0.001$.

Author's Response to Decision Letter for (RSOS-191220.R0)

See Appendix A.

RSOS-191220.R1 (Revision)

Review form: Reviewer 1

Is the manuscript scientifically sound in its present form?

No

Are the interpretations and conclusions justified by the results?

No

Is the language acceptable?

Yes

Do you have any ethical concerns with this paper?

No

Have you any concerns about statistical analyses in this paper?

No

Recommendation?

Major revision is needed (please make suggestions in comments)

Comments to the Author(s)

Authors, partially addressed my suggestions, but the article has some critical issues that needs to be considered.

The title "Fish-Robot Interaction in a Salmon Aquaculture Sea Cage" does not reflect what has been done in the work, and this may confuse readers.

I cannot see a fish-robot interaction in this study. This work is far from this scientific field.

Authors used a robot, not bioinspired, in Salmon Aquaculture Sea Cage to observe/monitor them. An interaction has not been established.

Please, rephrase the title, as well as the abstract and the introduction. This study is not the first in using robot in fish-robot interactions outside the laboratories. Your robot here is used to observe fish. Similar to your robot, previous studies carried out experiments with robots in the ocean to observe fish (e.g. Katzschmann, R. K., DelPreto, J., MacCurdy, R., & Rus, D. (2018). Exploration of underwater life with an acoustically controlled soft robotic fish. *Science Robotics*, 3(16), eaar3449).

Experimental conditions are still vague and not acceptable for a scientific publication.

Authors replies that "The average velocity changes depending on the position in the sea cage as well as the time of day and sea state", of course I should agree at the microenvironment level with them. But information about this concerning the local geographical features should be included in the text. I am sure these values are different in cages located in Norway form cages located in China. These details are important to ensure reproducibility. Your experiment produced these results in those environmental conditions. You do not need "...flow measurement device...", but probably these data are available from a local weather station.

Color measurements have not been provided. This is a fundamental aspect of your work. I provided several works in the previous review process, including the importance of colors in fish experiments with robots. There you can find how it is relevant and which methods have been used, but probably you considered them not relevant, it is ok, however, please address these important issues.

Authors replies that "... for the marine environment the spectral shift varies throughout the day, and is a function of the depth, ambient lighting as well as the properties of the water. Fish vision is indeed important, but the change in color in the subsea environment is based on several physical phenomena which would make a color measurement highly uncertain. For a recent overview of underwater imagery correction, we recommend: <https://iopscience.iop.org/article/10.1088/1757-899X/571/1/012125/meta>".

Although I have a solid knowledge on these aspects I would like to tanks Authors for recommending a recent overview <https://iopscience.iop.org/article/10.1088/1757-899X/571/1/012125/meta>.

However, I would like to point out to the authors that the real world (e.g. ecosystems), has infinite elements perturbing a system, and we cannot manage them completely. But this does not authorize a scientist to leave out important details of the system she/he developed. A scientists should be more accurate as possible to ensure reliability and reproducibility for other researchers. What use would this work have if nobody could reproduce it? What contribution would it have? What fish perceive in those conditions could be uncertain but if a researcher or a fish farmer would like to use this system probability a green robot would have different impact from a brown robot, although both are affected by physical phenomena in water. Maybe robots with the same color are affected more similarly than robots with different color, so you allow to closer reproduce your conditions.

The approach of not considering useful to report some (important) details information because there are environmental turbulences that could modify it is completely un-scientific, and I strongly encourage Authors to consider my suggestions to improve their work.

Review form: Reviewer 2

Is the manuscript scientifically sound in its present form?

Yes

Are the interpretations and conclusions justified by the results?

Yes

Is the language acceptable?

Yes

Do you have any ethical concerns with this paper?

No

Have you any concerns about statistical analyses in this paper?

No

Recommendation?

Accept as is

Comments to the Author(s)

The authors nicely revised their manuscript in accordance to the reviewers' suggestions and I recommend publication.

Decision letter (RSOS-191220.R1)

09-Dec-2019

Dear Dr Kruusmaa:

Manuscript ID RSOS-191220.R1 entitled "Fish-Robot Interaction in a Salmon Aquaculture Sea Cage" which you submitted to Royal Society Open Science, has been reviewed. The comments of the reviewer(s) are included at the bottom of this letter.

Please submit a copy of your revised paper before 01-Jan-2020. Please note that the revision deadline will expire at 00.00am on this date. If we do not hear from you within this time then it will be assumed that the paper has been withdrawn. In exceptional circumstances, extensions may be possible if agreed with the Editorial Office in advance. We do not allow multiple rounds of revision so we urge you to make every effort to fully address all of the comments at this stage. If deemed necessary by the Editors, your manuscript will be sent back to one or more of the original reviewers for assessment. If the original reviewers are not available we may invite new reviewers.

- Ethics statement

- Data accessibility

- Competing interests

- Authors' contributions

- Acknowledgements

- Funding statement

Kind regards,

Anita Kristiansen

Editorial Coordinator

on behalf of Dr Manoj Srinivasan (Associate Editor) and R. Kerry Rowe (Subject Editor)

Associate Editor Comments to Author (Dr Manoj Srinivasan):

Comments to the Author:

While the reviewers find merit in the paper, one of them has provided comments for revision. I agree with many of the comments and I think most of the comments can be addressed by adding appropriate caveats or stating the limitations of the study clearly. As the reviewer points out, while many of the experimental details were provided, some were not. You can either provide that information, or add a sentence or two that clearly lists all the potential confounds that you did not control for and state that perhaps that the results could be specific to those conditions (e.g., color, weather/microenvironment, etc.). This is only fair and would help a reader better judge the result or think about controlling those issues in a future study. Consider citing the articles suggested by the reviewer. For instance, it may be better to include a sentence about robot color in the article itself rather than respond to the reviewer through the 'response to reviewer' document, as other readers may have similar questions.

As the reviewer suggests consider changing the title to perhaps something like 'Effect of robot type on fish behavior in a Salmon Aquaculture Sea Cage' or some variant thereof? Please also consider citing the reference suggested by the reviewer.

Reviewer comments to Author:

Reviewer: 1

Comments to the Author(s)

Authors, partially addressed my suggestions, but the article has some critical issues that needs to be considered.

The title "Fish-Robot Interaction in a Salmon Aquaculture Sea Cage" does not reflect what has been done in the work, and this may confuse readers.

I cannot see a fish-robot interaction in this study. This work is far from this scientific field. Authors used a robot, not bioinspired, in Salmon Aquaculture Sea Cage to observe/monitor them. An interaction has not been established.

Please, rephrase the title, as well as the abstract and the introduction. This study is not the first in using robot in fish-robot interactions outside the laboratories. Your robot here is used to observe fish. Similar to your robot, previous studies carried out experiments with robots in the ocean to observe fish (e.g. Katzschmann, R. K., DelPreto, J., MacCurdy, R., & Rus, D. (2018). Exploration of underwater life with an acoustically controlled soft robotic fish. *Science Robotics*, 3(16), eaar3449).

Experimental conditions are still vague and not acceptable for a scientific publication.

Authors replies that "The average velocity changes depending on the position in the sea cage as well as the time of day and sea state", of course I should agree at the microenvironment level with them. But information about this concerning the local geographical features should be included in the text. I am sure these values are different in cages located in Norway form cages located in China. These details are important to ensure reproducibility. Your experiment produced these results in those environmental conditions.

You do not need "...flow measurement device...", but probably these data are available from a local weather station.

Color measurements have not been provided. This is a fundamental aspect of your work.

I provided several works in the previous review process, including the importance of colors in fish experiments with robots. There you can find how it is relevant and which methods have been used, but probably you considered them not relevant, it is ok, however, please address these important issues.

Authors replies that "... for the marine environment the spectral shift varies throughout the day, and is a function of the depth, ambient lighting as well as the properties of the water. Fish vision is indeed important, but the change in color in the subsea environment is based on several physical phenomena which would make a color measurement highly uncertain. For a recent overview of underwater imagery correction, we recommend: <https://iopscience.iop.org/article/10.1088/1757-899X/571/1/012125/meta>".

Although I have a solid knowledge on these aspects I would like to tanks Authors for recommending a recent overview <https://iopscience.iop.org/article/10.1088/1757-899X/571/1/012125/meta>.

However, I would like to point out to the authors that the real world (e.g. ecosystems), has infinite elements perturbing a system, and we cannot manage them completely. But this does not authorize a scientist to leave out important details of the system she/he developed. A scientists should be more accurate as possible to ensure reliability and reproducibility for other researchers. What use would this work have if nobody could reproduce it? What contribution would it have? What fish perceive in those conditions could be uncertain but if a researcher or a fish farmer would like to use this system probability a green robot would have different impact from a brown robot, although both are affected by physical phenomena in water. Maybe robots with the same color are affected more similarly than robots with different color, so you allow to closer reproduce your conditions.

The approach of not considering useful to report some (important) details information because there are environmental turbulences that could modify it is completely un-scientific, and I strongly encourage Authors to consider my suggestions to improve their work.

Reviewer: 2

Comments to the Author(s)

The authors nicely revised their manuscript in accordance to the reviewers' suggestions and I recommend publication.

Author's Response to Decision Letter for (RSOS-191220.R1)

See Appendix B.

Decision letter (RSOS-191220.R2)

06-Feb-2020

Dear Dr Kruusmaa,

It is a pleasure to accept your manuscript entitled "Salmon Behavioural Response to Robots in an Aquaculture Sea Cage" in its current form for publication in Royal Society Open Science. The comments of the reviewer(s) who reviewed your manuscript are included at the foot of this letter.

on behalf of Dr Manoj Srinivasan (Associate Editor) and R. Kerry Rowe (Subject Editor)
openscience@royalsociety.org

Appendix A

Dear Editor,

We are thankful for the opportunity to revise our manuscript for publication in Royal Society Open Science.

Detailed responses to the reviewers' comments are provided following the reviewer's comments in italics. In general, the reviewer comments provided critical insight and valuable suggestions to improve the manuscript. Many of these were related to improving the explanation of the statistical analysis, and for these concrete suggestions we are especially thankful. We believe that after significant revision, we have been able to successfully address all comments in full. The only exception to our comprehensive revisions are related to Reviewer 1, who requested the additional citation of some 10 papers. After careful review of the suggested citations, we have added only those to the revised manuscript which we found to be substantially helpful. Regarding language improvements, we have re-written large portions of the text to improve clarity, which are marked in red to distinguish the previous and current versions of the manuscript.

On behalf of the co-authors and myself,

Prof. Maarja Kruusmaa

Reviewers' Comments to Author:

Reviewer: 1

Also relevant literature has not been reported.

Pag.3 line 2: Please include these recent reviews on animal-robot interactions:

*Romano, D., Donati, E., Benelli, G., & Stefanini, C. (2019). A review on animal–robot interaction: from bio-hybrid organisms to mixed societies. *Biological cybernetics*, 113(3), 201-225.*

*Porfiri, M. (2018). Inferring causal relationships in zebrafish-robot interactions through transfer entropy: A small lure to catch a big fish. *Animal Behavior and Cognition*, 5(4), 341-367.*

Pag.3 line 11: Also, robots were used in predator-prey interactions, as well as to encode desired behaviour in insects of economic importance. Please include these relevant works in your article:

*Cianca, V., Bartolini, T., Porfiri, M., & Macri, S. (2013). A robotics-based behavioral paradigm to measure anxiety-related responses in zebrafish. *PLoS One*, 8(7), e69661.*

*Ladu, F., Bartolini, T., Panitz, S. G., Chiarotti, F., Butail, S., Macri, S., & Porfiri, M. (2015). Live predators, robots, and computer-animated images elicit differential avoidance responses in zebrafish. *Zebrafish*, 12(3), 205-214.*

*El Khoury, R., Ventura, R. B., Cord-Cruz, G., Ruberto, T., & Porfiri, M. (2018, March). Interactive experiments in a robotics-based platform to simulate zebrafish response to a predator. In *Bioinspiration, Biomimetics, and Bioreplication VIII* (Vol. 10593, p. 1059301). International Society for Optics and Photonics.*

*Romano, D., Benelli, G., & Stefanini, C. (2017). Escape and surveillance asymmetries in locusts exposed to a Guinea fowl-mimicking robot predator. *Scientific Reports*, 7(1), 12825.*

*Romano, D., Benelli, G., & Stefanini, C. (2019). Encoding lateralization of jump kinematics and eye use in a locust via bio-robotic artifacts. *Journal of Experimental Biology*, 222(2), jeb187427.*

Pag.3 line 16: closed-loop control were used to enhance the degree of biomimicry of a robotic fish to interact with zebrafish:

*Kim, C., Ruberto, T., Phamduy, P., & Porfiri, M. (2018). Closed-loop control of zebrafish behaviour in three dimensions using a robotic stimulus. *Scientific reports*, 8(1), 657.*

*Pag.3 line 20: Robots were used to investigate and modulate the aggressive and courthship behaviours of a highly territorial species such as Siamese fighting fish to multimodals cues including light, an inedited signal in this fish:Romano, D., Benelli, G., Donati, E., Remorini, D., Canale, A., & Stefanini, C. (2017). Multiple cues produced by a robotic fish modulate aggressive behaviour in Siamese fighting fishes. Scientific reports, 7(1), 4667.
Romano, D., Benelli, G., Hwang, J. S., & Stefanini, C. (2019). Fighting fish love robots: mate discrimination in males of a highly territorial fish by using female-mimicking robotic cues. Hydrobiologia, 833(1), 185-196.*

After careful review of each of the suggested additional citations, we have included some of the above citations were we believe the reader would clearly benefit from their addition. As the reviewer correctly points out, the body of literature on animal-robot interaction is large and growing, and we wish to direct the reader specifically to sources which are germane to the revised manuscript.

Pag.3 line 36: Please, tone down this statement. You are carrying out an experiment where you use a biometric artifcat to observe in the field fish without disturbing them. This is far from a real ecological interaction(e.g. social, courtship, agonistic, predatation, parasitic, etc.). In addition, please clarify that robots you are using are not biomimetic, so they are not intended at to be perceived as a conspecific in the group of animals but it has just bioinspired features mainly focused on the robots locomotion.

We concur with the reviewer. The phrase “bio-mimetic robot” and similar is now replaced with the more appropriate term related only to actuation, as a “flipper-propelled robot”. In the revised manuscript we no longer state the claim that we replicate fish or other animals whatsoever.

Pag.3 line 37-pag.4 line 5: Please, reduce this part describing what is a fish farm and what are its potentials. It decreases the readability of the manuscript.

As the revised manuscript title reinforces, we find the focus on salmon aquaculture to be a salient feature of this work. By clarifying the experimental environment, the revised manuscript has improved the clarity of the work’s objective and significance for using underwater robots in real-world applications. We have substantially revised the body text of the manuscript, and believe the current version includes a minimal description of the rationale and methodology of the field experiments, which are necessary for future researchers to replicate our major findings, and for potential end-users to see the value of a practical method to evaluate fish behavior using underwater robots in aquaculture sea pens.

Pag. 4 lines 23-28: Put this part in "Materials and methods" section.

Thank you for the suggestion, we have moved the text to Materials and Methods to improve readability.

Pag.5 line 49-50: Please, provide more rigorous and detailed information on the experimental conditions.

what was the average velocity of water current?

what time of the day was selected to carry out tests?

If you have some information about light conditions would be greatly beneficial for your study.

how long did the test last?

How many tests were performed?

The average velocity changes depending on the position in the sea cage as well as the time of day and sea state. Because these variables are not practical to measure in field settings with live animals (e.g. a flow measurement device would disturb the fish), it was determined early on in the experimental design stage that currents and waves could not be included. We have specified the other details, regarding the duration of tests and number of tests.

Pag. 6 line 28: Inspired to which animal? This is important to report in an animal-robot experiment. Also, is important to clarify that your robot is not reproducing features of the animal model studied.

The revised manuscript has been corrected, and we now refer to U-CAT as a “flipper-propelled robot” without any reference to bio-inspiration.

Pag. 6 lines 45-49: Please, provide color measurements for each robot. This could importantly affect the response of fish. Why was the color different for each tested robot? How can you dissect the effect of color (vision is one of the most important sense in fish), of different robots from the effect produced by other features considered?

This is a relevant point, which is now discussed in more detail, including post-hoc testing in the subsection “Fish tailbeat frequency in reaction to U-CAT locomotion modes”, starting from page 12, line 11. We did not carry out color measurements, this is because for the marine environment the spectral shift varies throughout the day, and is a function of the depth, ambient lighting as well as the properties of the water. Fish vision is indeed important, but the change in color in the subsea environment is based on several physical phenomena which would make a color measurement highly uncertain. For a recent overview of underwater imagery correction, we recommend:

<https://iopscience.iop.org/article/10.1088/1757-899X/571/1/012125/meta>

Pag. 9 line 11: How was the distribution of your data? Maybe a non-parametric approach would be more appropriate.

The mean behavior data were Gaussian, and the only notable differences were considering the body-mounted GoPro evaluation imagery and the silver U-CAT body color, where it was observed that the number of outliers increased. For this reason, we did not opt for non-parametric approaches. However, it is possible that the inclusion of more detailed behavioral metrics, beyond the distance and tail beat frequency could very well warrant the use of non-parametric statistical analysis.

Furthermore, a description of the experiment with animals has not been provided and just vague descriptions have been included.

A description is included in the Materials and Methods section, from page 5, line 11. Due to the paucity of experimental literature with large fish aggregates, we tried to follow the example of previous works (cited from page 2, lines 28-45). If the reviewer has suggestions for specific descriptors, we would endeavor to include them in a future revision as necessary.

A careful English revision is needed.

The paper has been substantially revised. The updated text is marked in red to distinguish it from the previous version.

Reviewer: 2

Comments to the Author(s)

The manuscript „Underwater Robot Interaction with a Large Fish Aggregation” by Kruusmaa investigated live fish responses in an aquaculture facility towards moveable objects like divers and robots. As pointed out by the authors, any robot-fish interactions are investigated in the lab and more field-based analysis on the possible application of robots are needed. The results of the current paper show that a smaller, flipper-propelled robot (U-CAT) elicits less avoidance reactions of the fish compared to a human diver and another, bigger thruster-propelled robot. This is a very interesting study and the acquired knowledge will for sure foster new developments in the field of bioinspired robots as well as robot-animal interaction. However, I feel that the manuscript cannot be accepted for publication in its current form as the statistical analyses are not presented at a sufficiently high scientific level. For example, only p-values are provided with no information whether they stem from the global ANOVA models or subsequent posthoc tests. No F-Values or other properties of the ANOVA results are shown. This makes it hard to judge whether the effects are really there or not. Furthermore, I could not find a clear statement on the sample size except an N=50 for all ANOVAS and a 50 GB video file size. What does N=50 mean in the respective cases? Number of analyzed video snapshots or trial runs? How was the sample size distributed among treatments? equally? Also, figure legend are way to minimalistic to full understand what is shown (Median or mean? What do the box and whiskers mean?). I strongly recommend putting more details into the description of the statistics!

Much of the confusion of the review arose from that we did not mention that N=50 is the number of individuals measured. To improve clarity, we have now emphasized this point, removing most of the confusion. All figures with statistical analysis now contain updated captions explaining how they should be read. We have also performed Fisher test and Tukey-Kramer post-hoc test on datasets, where appropriate. We thank the reviewer for their concrete suggestions on improving the use of statistics, and believe the revised manuscript is substantially improved.

Specific comments:

P2L46: Sentence “At the same time...” is not needed and can be deleted

Removed as suggested.

P3L14: Golden Shiners

Updated.

P3L50 food instead of feed?

Amended accordingly.

P3L57: Better connection from environmental variation to robots need.

This section has been substantially revised.

P6L40: Please be consistent: either report length as cm or m

All lengths reported in meters.

P7L12: Table legend: More details are needed to understand that table. What do plus and minus mean?

The table caption has been updated accordingly.

P8L35: 350 GB: Minutes of observation would be a much better proxy for the amount of raw data acquired as 4K videos often have 2 GB per minute or even more but this highly depends on the camera model used.

The manuscript is updated now reporting that the total amount of videos was 24 h and post-processed videos 7,1 hours.

P9L1: What does that mean "converged towards the mean value at around 25-30 fish encounters?" In addition, N=50, is this the number of experimental runs in total (when considering one run as a 5 min sequence that was described before above) or is it 50 analyzed pics? per treatment? in total? Please clarify.

Additional detail provided, please see page 8, lines 2-6.

P9L8: Repeatability analysis can be put into the supplement.

We would prefer to keep this in the main text, as we believe it is an important feature of large-scale study using field data.

P10L37: Please provide statistical evidence (model values like F-values, mean squares, etc).

We appreciate this specific comment, and have included F-values and post-hoc tests where appropriate to improve the description of the statistical analysis.

P10L44: P=0.04: The authors mention a small difference but also a significantly small P-value. Please clarify.

We have significantly revised the text outlining the statistical analysis and results to improve clarity and readability.

P11L21: Figure 4 legend: Please tell the sample size of analyzed frames and test runs per treatment. Tell what the box and the whiskers mean.

Additional text has been added to all box and whisker plot captions to improve clarity.

P11L26: "Analysis of the distance..." This sentence is incorrect. Fig. 5 does not show any analysis. It shows data!

We concur, the text has been amended accordingly.

P12L30: The description of the ANOVA results reads strange. Please revise.

Revised as suggested.

P12L8: P-values smaller than 0.001 are reported as P<0.001.

We thank the reviewer for this correction, we have now

Appendix B

Dear Editor,

We would like to thank the two anonymous reviewers for their focused, insightful and helpful suggestions on improving this manuscript. In the following written response to comments, we have replied to the editor's as well as the first reviewer's comments. Our responses are marked in blue text immediately following the associated comment. There were no further comments from the second reviewer to address.

We believe that we were able to address all comments in the second revision of the manuscript, which has been substantially improved by the review process.

Prof. Maarja Kruusmaa
On behalf of myself and the co-authors

Comments to Author: Editor's Remarks:

Associate Editor Comments to Author (Dr Manoj Srinivasan):
Comments to the Author:

While the reviewers find merit in the paper, one of them has provided comments for revision. I agree with many of the comments and I think most of the comments can be addressed by adding appropriate caveats or stating the limitations of the study clearly. As the reviewer points out, while many of the experimental details were provided, some were not. You can either provide that information, or add a sentence or two that clearly lists all the potential confounds that you did not control for and state that perhaps that the results could be specific to those conditions (e.g., color, weather/microenvironment, etc.). This is only fair and would help a reader better judge the result or think about controlling those issues in a future study. Consider citing the articles suggested by the reviewer. For instance, it may be better to include a sentence about robot color in the article itself rather than respond to the reviewer through the 'response to reviewer' document, as other readers may have similar questions.

We agree with most of the comments from the first reviewer. The difficulty of documenting many of the experimental details is as the first reviewer points out, that there are "infinite degrees of freedom" in the natural environment. Our experiments were indeed planned with both local academic and industrial experts, and were purposely focused on obtaining significant, relevant research results applicable to future, full-scale aquaculture studies. We therefore understand the critique that highly detailed background information was not included, but assert that this is not a weakness of the work, but rather keeps the research focus on a real-world implementation of how underwater robots and a human diver may create different outcomes when monitoring aquaculture facilities.

In consideration of the editor and the first reviewer's concerns, we have changed the title of the paper, edited the introduction to better reflect the nature of the work, and amended the body text accordingly to make it clear which confounding factors may exist, and how future research may seek to look into more detailed background effects, where possible. An additional table (Table 1 in the revised manuscript) has now been included which provides daily statistics for both the weather conditions as well as the tidal levels.

As the reviewer suggests consider changing the title to perhaps something like 'Effect of robot type on fish behavior in a Salmon Aquaculture Sea Cage' or some variant thereof? Please also consider citing the reference suggested by the reviewer.

We agree that the new title could be improved. It has been changed to "Salmon Behavioural Response to Robots in an Aquaculture Sea Cage".

Reviewer comments to Author:

Reviewer: 1

Comments to the Author(s)

Authors, partially addressed my suggestions, but the article has some critical issues that needs to be considered.

The title “Fish-Robot Interaction in a Salmon Aquaculture Sea Cage” does not reflect what has been done in the work, and this may confuse readers.

I cannot see a fish-robot interaction in this study. This work is far from this scientific field. Authors used a robot, not bioinspired, in Salmon Aquaculture Sea Cage to observe/monitor them. An interaction has not been established.

We have changed the title of the paper to “Salmon Behavioural Response to Robots in an Aquaculture Sea Cage” in response to the reviewer’s concerns.

Please, rephrase the title, as well as the abstract and the introduction. This study is not the first in using robot in fish-robot interactions outside the laboratories. Your robot here is used to observe fish. Similar to your robot, previous studies carried out experiments with robots in the ocean to observe fish (e.g. Katzschmann, R. K., DelPreto, J., MacCurdy, R., & Rus, D. (2018). Exploration of underwater life with an acoustically controlled soft robotic fish. *Science Robotics*, 3(16), eaar3449).

Experimental conditions are still vague and not acceptable for a scientific publication.

Authors replies that “The average velocity changes depending on the position in the sea cage as well as the time of day and sea state”, of course I should agree at the microenvironment level with them. But information about this concerning the local geographical features should be included in the text. I am sure these values are different in cages located in Norway form cages located in China. These details are important to ensure reproducibility. Your experiment produced these results in those environmental conditions.

Detailed information on the site-specific conditions have now been added including the fjord dimensions and typical flow conditions faced at the site. In addition, we have received information from SINTEF Ocean (personal communication, Mr. Kevin Frank) regarding the currents in the vicinity of the farm site during our investigation which are now included in an expanded site description at the beginning of the Materials and Methods section.

You do not need “...flow measurement device...”, but probably these data are available from a local weather station.

We have included a new table which includes the daily weather statistics as well as the tidal levels.

Color measurements have not been provided. This is a fundamental aspect of your work. I provided several works in the previous review process, including the importance of colors in fish experiments with robots. There you can find how it is relevant and which methods have been used, but probably you considered them not relevant, it is ok, however, please address these important issues.

Color information based on digital image analysis during field conditions has been provided. The reviewer is correct in stating that color may be an important factor, however the literature presented did not provide a detailed methodology which we found to be practical for field conditions to assess color. For this reason, we have applied a simple image-based analysis method which others can also apply in future studies.

Authors replies that "... for the marine environment the spectral shift varies throughout the day, and is a function of the depth, ambient lighting as well as the properties of the water. Fish vision is indeed important, but the change in color in the subsea environment is based on several physical phenomena which would make a color measurement highly uncertain. For a recent overview of underwater imagery correction, we recommend: <https://iopscience.iop.org/article/10.1088/1757-899X/571/1/012125/meta>".

Although I have a solid knowledge on these aspects I would like to tanks Authors for recommending a recent overview <https://iopscience.iop.org/article/10.1088/1757-899X/571/1/012125/meta>.

However, I would like to point out to the authors that the real world (e.g. ecosystems), has infinite elements perturbing a system, and we cannot manage them completely. But this does not authorize a scientist to leave out important details of the system she/he developed. A scientists should be more accurate as possible to ensure reliability and reproducibility for other researchers. What use would this work have if nobody could reproduce it? What contribution would it have?

The reviewer's remarks are understandable. However, we believe that reproducibility of experiments taken in the outdoors with thousands of fish in an aquaculture sea cage is always going to pose a significant challenge. To deal with this reality, we have tried to focus our field work and methods to provide a robust, practical and statistically sound assessment of salmon monitoring in an aquaculture sea cage. Should follow-up experiments prove possible, we will allocate additional effort in the recording of possible cofounding factors related to environmental and operational conditions (e.g. water temperature, salinity, light, underwater sound using hydrophone).

What fish perceive in those conditions could be uncertain but if a researcher or a fish farmer would like to use this system probability a green robot would have different impact from a brown robot, although both are affected by physical phenomena in water. Maybe robots with the same color are affected more similarly than robots with different color, so you allow to closer reproduce your conditions.

We understand the reviewer's concern regarding the colors associated with the devices. To address this, we have added the analysis of the colors based on images taken of the diver, U-CAT and Argus Mini during different lighting conditions, and from different viewpoints. We agree with the reviewer's concerns, and believe it was important to include this information based on what we believe to be a robust and simple method which others can apply in the field for future studies. Should future follow-up studies prove feasible, we will also include an analysis of the underwater environment's impact on body color and its effect on fish using standard color charts and the fish monitoring method proposed in this work.

The approach of not considering useful to report some (important) details information because there are environmental turbulences that could modify it is completely un-scientific, and I strongly encourage Authors to consider my suggestions to improve their work.

We appreciate the reviewer's critical comments, and believe we have addressed them in full.